# Centrality in the host–pathogen interactome is associated with pathogen fitness during infection

Núria Crua Asensio[1], Elisabet Muñoz Giner[1], Natalia Sánchez de Groot[2,3] & Marc Torrent Burgas[1,4]

To perform their functions proteins must interact with each other, but how these interactions influence bacterial infection remains elusive. Here we demonstrate that connectivity in the host–pathogen interactome is directly related to pathogen fitness during infection. Using *Y. pestis* as a model organism, we show that the centrality-lethality rule holds for pathogen fitness during infection but only when the host–pathogen interactome is considered. Our results suggest that the importance of pathogen proteins during infection is directly related to their number of interactions with the host. We also show that pathogen proteins causing an extensive rewiring of the host interactome have a higher impact in pathogen fitness during infection. Hence, we conclude that hubs in the host–pathogen interactome should be explored as promising targets for antimicrobial drug design.

[1] Systems Biology of Infection Lab, Department of Microbiology, Vall d'Hebron Institut de Recerca (VHIR), Passeig Vall d'Hebron 119-129, 08035 Barcelona, Spain. [2] Gene Function and Evolution Lab, Centre for Genomic Regulation (CRG), Dr Aiguader 88, 08003 Barcelona, Spain. [3] Universitat Pompeu Fabra (UPF), Dr. Aiguader 88, 08003 Barcelona, Spain. [4] Universitat Autònoma de Barcelona, Department of Biochemistry and Molecular Biology, Biosciences Faculty, 08193 Cerdanyola del Vallès, Spain. Correspondence and requests for materials should be addressed to N.S.d.G. (email: natalia.sanchez@crg.eu) or to M.T.B. (email: marc.torrent@vhir.org).

Pathogens tend to interact with hubs and bottlenecks, that is, those with high degree and centrality in the human protein-protein interaction (PPI) network[1,2]. In this scenario, bacteria may have adapted to attack proteins involved in specific pathways, most importantly in immunity and defence mechanisms[3,4]. However, whether protein–protein interactions with the host are correlated to pathogen fitness during infection is currently unknown.

Highly connected nodes in the protein network of an organism (hubs) tend to be essential, which is a property known as the centrality-lethality rule[5,6]. This rule holds for many organisms, both prokaryotes and eukaryotes[5,7,8]. However, the extension of this rule to infectious organisms such as bacteria is not straightforward. Infectious bacteria, in contrast to well-studied unicellular eukaryotes (for example, *Saccharomyces cerevisiae*) or higher multicellular organisms (for example, *Drosophila melanogaster*), require a host to grow and reproduce in their natural environment. Thus, the host imposes the definition of essentiality in pathogenic bacteria, and the fitness of a bacterium is directly related to a successful interaction with the host. In this paper, we aim to answer whether protein connectivity in the host–pathogen interactome is related to pathogen fitness during infection.

## Results

**Hubs in the interactome are central for pathogen fitness.** We examined both the bacteria and host–bacteria interactomes of *Yersinia pestis* (see Methods and Supplementary Methods sections for a complete description). For each protein, we correlated the network topological information with pathogen fitness data (that is, the cost of deleting a protein for the pathogen to grow) measured either in rich media (*in vitro*) or after host infection (*in vivo*)[9]. First, we analysed how the degree of connection in the pathogen interactome (without the presence of the host) correlates with its fitness both *in vitro* and *in vivo*. We determined that *Y. pestis* follows the centrality-lethality rule *in vitro* ($r^2 = 0.85$, $P = 0.0030$, Fig. 1a), as observed in other prokaryotes[7], but not *in vivo* ($r^2 = 0.38$, $P = 0.19$, Fig. 1a). These data lead to the intriguing conclusion that the network structure of the bacterial proteome is unrelated to its infection capacity. On the contrary, we observed a significant positive trend between the node degree in the *Homo sapiens–Y. pestis* interactome and pathogen fitness *in vivo* ($r^2 = 0.81$, $P = 0.0060$, Fig. 1b) but not *in vitro* ($r^2 = 0.38$, $P = 0.19$, Fig. 1b). In summary, our results suggest that highly connected nodes in the host–pathogen interactome are more important for infection than less-connected nodes. Most interestingly, we also found similar correlations in other bacterial pathogens (*Salmonella enterica* and *Acinetobacter baumanii*; Supplementary Fig. 1) suggesting that our observations are robust and can be generalized to other bacterial pathogens. Hence, we conclude that the centrality-lethality rule holds for pathogen fitness during infection only when the host–pathogen interactome is considered.

**Connectivity explains *in vivo* and *in vitro* fitness changes.** When we plotted the fitness effect of deleting a certain protein *in vitro* against its effect *in vivo*, we observed a negative correlation (Fig. 2a) meaning that deletions resulting in severe growth reduction *in vivo* have only mild effects *in vitro* and vice versa (for example, the pesticin receptor *fyuA* or the phospho-carrier protein *ptsO* are required for survival in the host but dispensable in culture).

Although a large fraction of proteins do not influence pathogen fitness (∼60%; grey data points displayed in Fig. 2a) we could define two groups that have a high impact either *in vitro*

or *in vivo* when deleted (orange and red data points in Fig. 2a, respectively). Based on our previous observations, we reasoned that this behaviour could be related to their centrality in the interactome. To test this hypothesis, we measured the degree of proteins belonging to these three groups both in the pathogen and host–pathogen interactomes. We found that proteins with a higher effect *in vitro* have a larger number of contacts in the pathogen interactome than the other two groups (left panel, Fig. 2b). Conversely, genes with higher effect *in vivo* showed a larger number of contacts in the host-pathogen interactome (right panel, Fig. 2b).

Intriguingly, we could not find any particular biological function associated to any of these groups, suggesting that these proteins are not specifically related to macromolecular complexes or defined signalling pathways. Hence, we reasoned that the critical impact of these proteins in pathogen survival might be related to the functions of their interacting partners. First, we examined the host proteins that interact with pathogen proteins relevant for infection (red data points). We found a significant enrichment in functions related to the immune response, transcription regulation and vesicle transport (Fig. 2c). These functions are highly relevant for the pathogenesis of *Y. pestis* because allow the pathogen to survive inside macrophages during the early stages of infection[10] and contribute to pathogen survival[11]. In the case of interactions relevant for pathogen growth in culture (orange data points), we found a wider collection of functions, most of them related to cell metabolism and DNA homeostasis (Fig. 2d), which are indeed fundamental in conditions of maximal microbial proliferation.

Based on the results presented here, it is tempting to hypothesize that the nature of pathogen–pathogen and host–pathogen interactions is fundamentally different, even orthogonal. In this context, pathogens might have evolved a subset of the proteome to make specific interactions with the host, restricting their number of interactions within the pathogen. These proteins, though fundamental for host infection, would become more isolated from the pathogen network, explaining why they are mostly dispensable in a non-host environment.

**Pathogens target but not disrupt host networks.** As mentioned before, pathogen proteins bind to specific targets in the host that are involved in defined biological functions (A summary of all significant biological functions found for *Y. pestis* can be found in Supplementary Fig. 2). To analyse how the networks associated to these functions are affected during infection we simulated a network attack based on pathogen-directed interactions and compared the results with random and centrality-based attacks. To illustrate the strategy used we present the endocytosis function as an exemplar in Fig. 3a. We found that a pathogen-directed attack decreases the global topological efficiency of the network significantly more than a random attack but much less than a centrality-directed approach (Fig. 3b). These results indicate that, though pathogens tend to interact with hubs in the host network to hijack it for their own profit[1], they do not completely disrupt its integrity, probably because it would cause a catastrophic failure of the host cell metabolism that would hinder bacteria survival and proliferation.

We also observed that certain host networks are more disturbed than others after a pathogen-based attack (Supplementary Fig. 2). We reasoned that this might have a relation with the importance of a particular network for pathogen survival and reproduction inside the host. To test this hypothesis, we plotted the decrease in global topological efficiency after a pathogen-based attack against the average fitness cost of

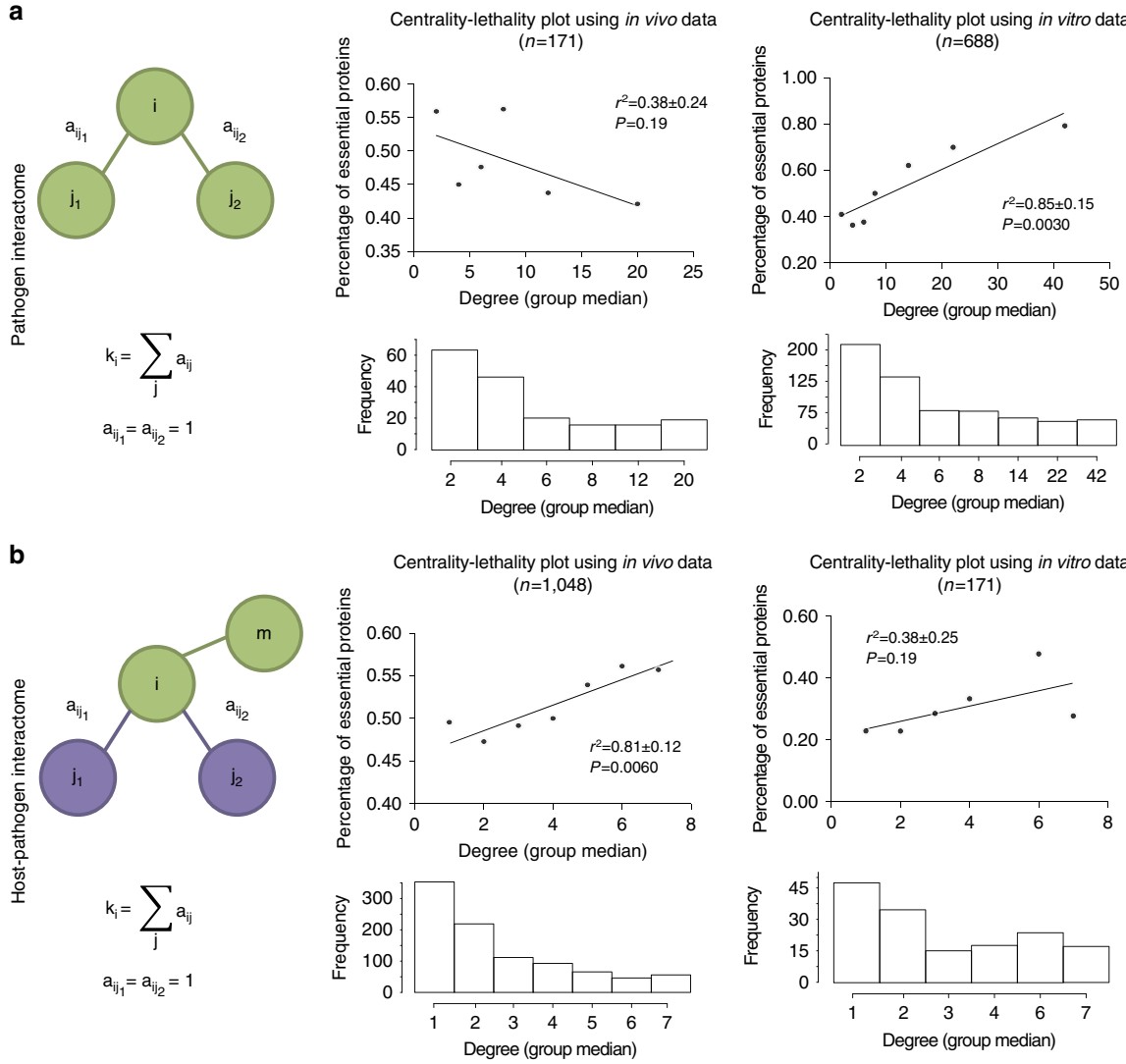

**Figure 1 | Analysis of the centrality-lethality rule in *Yersinia pestis*.** The percentage of essential proteins both *in vivo* (during infection) and *in vitro* (growth in culture medium) was plotted against its degree in (**a**) the pathogen (green) and (**b**) the host–pathogen interactome (purple). The total number of observations ($n$) is included in each graph ($n = 1,048$ for the *Y. pestis* interactome, $n = 688$ for the *H. sapiens*–*Y. pestis* interactome and $n = 171$ for proteins with data available in both interactomes). The degree ($k$) of a node $i$ is defined as the number of edges linked to $i$ as displayed in the figure. To control the effect size we grouped the data in bins according to degree to ensure a comparable number of observations in each bin. Histograms below each graph indicate the number of observations included in each bin. $P$-values were calculated using the $t$-test and the confidence interval for $r^2$ was estimated by bootstrapping before binning the data as detailed in the Methods section.

deleting the pathogen proteins involved in the attack. The significant correlation found ($r^2 = 0.31$, $P = 0.032$, Fig. 3c) suggests not only that pathogens have evolved proteins to target host networks that critically affect its fitness but also that proteins that disrupt host networks are more essential for infection.

## Discussion
The availability of high-throughput technological advances together with powerful analytical tools developed in the field of Systems Biology allow to study the host and pathogen as a whole system instead of isolated entities. In the present study, we combined random yeast-two-hybrid screening and transposon-sequencing data to interrogate full interactomes with the aim to understand the relevance of protein–protein interactions in infectious diseases.

Our results show that pathogen proteins that make a higher number of interactions with the host also have a major impact in

the fitness of the organism during infection. In other words, we provide evidence that pathogen proteins have an impact in the outcome of infection that is proportional to its ability to reorganize the host interactome.

As in all large-scale studies, we should be aware of some limitations that are intrinsic to the methodologies used. High-throughput analysis of interactomes by yeast-two-hybrid screening can generate: (i) false negatives (that is, protein–protein interactions not detected due to limitations of the screening method) and (ii) false positives (that is, interactions detected in the screening that cannot be reproduced using an independent method). Despite these limitations, we found that a moderate degree of noise in the interactome does not invalidate our observations (Supplementary Methods). Another important caveat is whether these interactions actually occur during infection. For two proteins to interact they must be present in the same location at the same time. Unfortunately, the cellular and subcellular location of all these protein pairs during infection

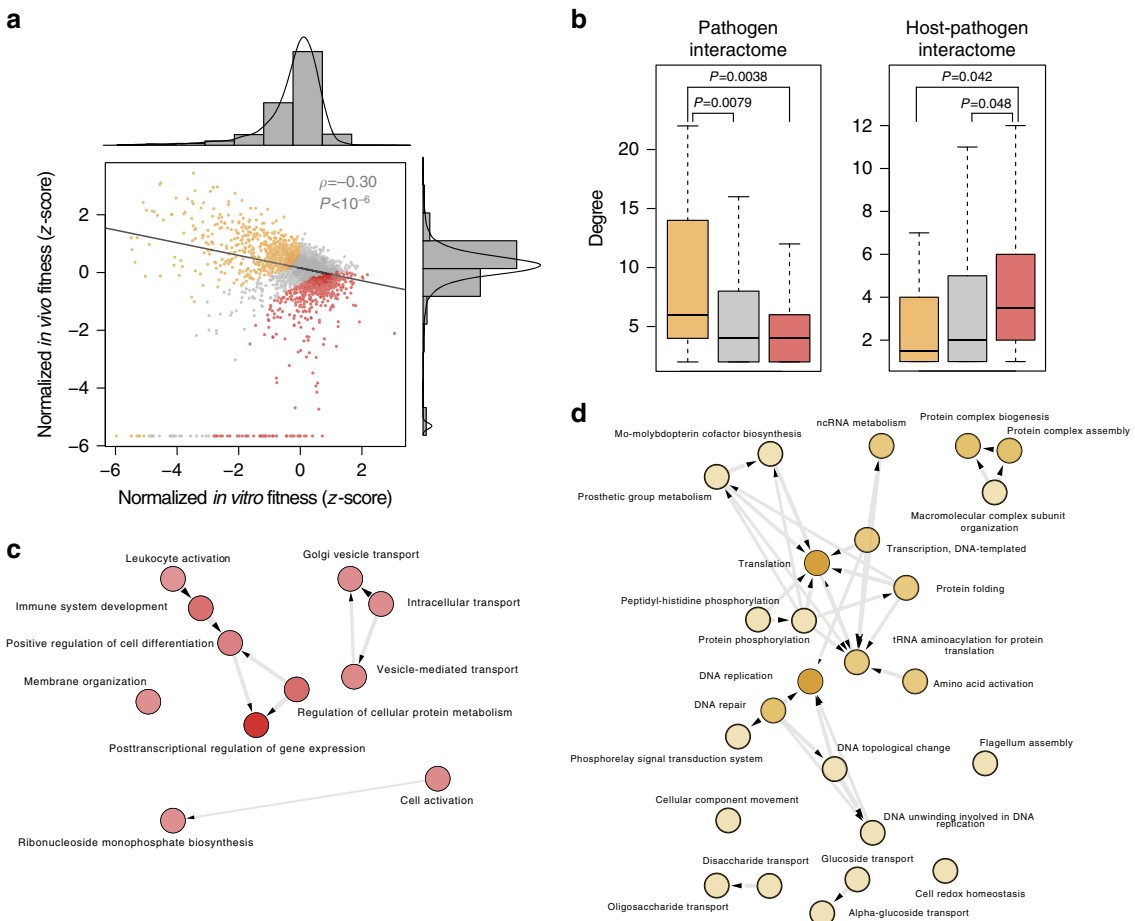

**Figure 2 | Analysis of fitness shifting during *Y. pestis* infection. (a)** Representation of *in vivo* against *in vitro* fitness z-scores (calculated according to $z=(x-\mu)/\sigma$). Proteins that rank 2.5 times higher *in vivo* than *in vitro* are coloured red while those ranking 2.5 times lower are coloured orange. The remaining proteins (that is, those whose rank does not change) are coloured grey. Spearman rank order correlation is shown. **(b)** Boxplots showing the average node degree in the pathogen (left) and host–pathogen (right) interactome for the three aforementioned categories. P-values were calculated using the Mann–Whitney U-test. Changing the threshold in the definition of orange, grey and red groups did not have a significant effect (for 2 < threshold < 3 calculated P-values were always significant). GO molecular function enrichment is plotted for interactors in the red **(c)** and orange **(d)** groups. For the red group, enrichment was calculated over the complete list of host interactors to avoid potential biases while for the orange group it was done over the complete *Y. pestis* proteome. GO categories were grouped using Revigo and plotted with Cytoscape as described in the Methods section. In box plots, the bottom and top represent the 25th and 75th percentile (i.e. lower and upper quartiles) and the centre line represents the median. The end whiskers are calculated according to the following: upper whisker = min(max(x), $Q_3 + 1.5 \times IQR$); lower whisker = max(min(x), $Q_1 - 1.5 \times IQR$).

has not been systematically investigated. Hence, it will be important in the future to identify when and where are these pathogen proteins delivered to host cells. New advances in the fields of genomics and proteomics, including dual-transcriptome sequencing[12] and crosslinking mass spectrometry[13] will provide more data in the near future that may help us to cope with these limitations.

Finally, our observations have important implications in drug design[14–16]. Antimicrobial development has focused on essential proteins (mainly enzymes) required for the pathogen to survive in culture (that is, *in vitro*). Although this approach has been successful in the past, recent antimicrobial resistance threatens our capacity to develop new drugs. Our results suggest that strategic protein–protein interactions in the host–pathogen interactome should be explored as putative drug targets that may lay the foundation of a new class of antimicrobials.

## Methods

**Databases.** *Yersinia pestis* was selected as a model organism because the host–pathogen interactome was thoroughly described and fitness datasets were available both *in vivo* and *in vitro*. When possible, studies were complemented with

data on *Salmonella enterica*[17] *and Acinetobacter baumanii*[18], two relevant human pathogens.

The *Y. pestis*, *S. enterica* and *A. baumanii* interactomes were obtained from the String database[19]; only experimentally validated interactions were included. The *H. sapiens*-*Y. pestis* interactome was downloaded from IntAct[20] as reported in ref. 4 and contains 4,059 human–*Y. pestis* protein–protein interactions from a random yeast-two-hybrid assay[4] with a tenfold coverage of the coding capacity of *Y. pestis*. All other host–pathogen interactomes (that is, for *S. enterica* and *A. baumanii*) were obtained by homology search using the HPIDB database[21]. Interactomes predicted by homology search may represent an incomplete version of the host–pathogen interactome, particularly in the case of rapidly evolving virulence factors[22]. Despite this fact, the number of observations was large enough for validation purposes (Supplementary Fig. 1). *Y. pestis* fitness data measured in rich media (*in vitro*) and after infection (*in vivo*) were obtained from ref. 9. Fitness data was calculated from transposon-sequencing (Tn-seq) data defined as the ratio of the rates of population expansion for the two genotypes[9]. In all, 1.5 million of independent insertion mutants were screened with a coverage of ~70% of the *Y. pestis* genome. *S. enterica* (10,368 mutants screened by whole-genome tiling microarrays) and *A. baumanii* (150,000 mutants screened by Tn-seq) fitness data were obtained from ref. 17 and ref. 18, respectively. We considered that proteins were essential for infection if their deletion promotes a pathogen fitness decrease below the median.

**Statistical analyses.** Unless otherwise specified all P-values were calculated using the t-test or Mann–Whitney U-test and considered significant when $P < 0.05$

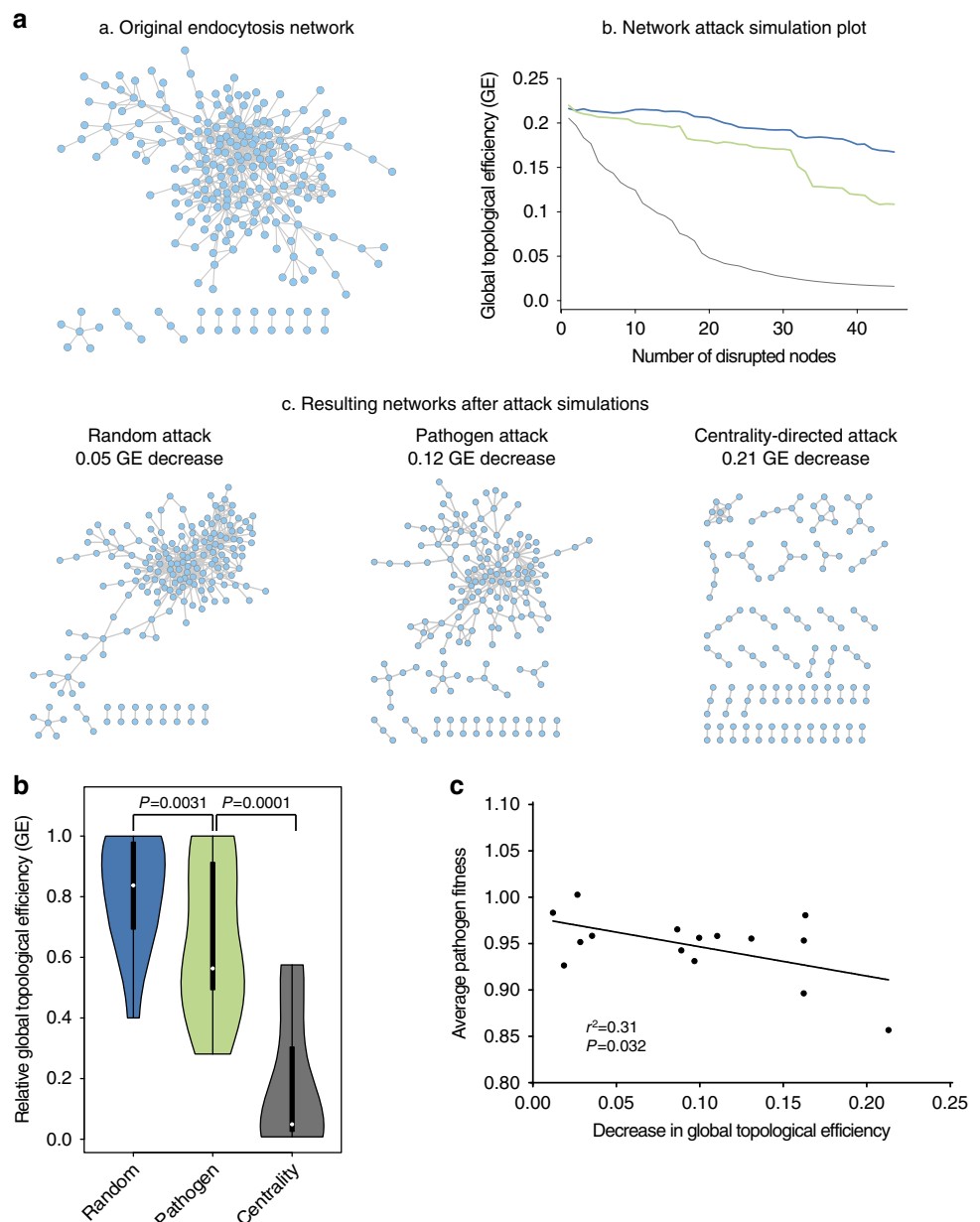

**Figure 3 | Simulation of the host network attack by the pathogen _Y. pestis._** In **a** we present an exemplar of the three attack strategies used, in this case for the endocytosis pathway. The endocytosis network (**a**) was subjected to successive node removal according to a random, pathogen or centrality-based attacks (**b**). The resulting networks after each attack (**c**) are plotted for each strategy. (**b**) Global topological efficiency (GE) decrease was measured in different host networks after a random attack (blue, average of 20 simulations), a pathogen-directed attack (green) and a centrality-based attack (black). _P_-values were calculated using the Mann–Whitney _U_-test. (**c**) GE decrease was plotted against the average fitness cost of deleting the pathogen proteins involved in the attack.

(see Figure Legends for further details). Bootstrapping analyses were conducted in R using the boot package. In all cases, 10,000 resampling cycles with replacement were obtained and 95% confidence intervals were calculated using the adjusted bootstrap percentile interval method[23].

**Calculation of network parameters.** All protein networks were analysed with Cytoscape and statistical calculations were done in R. The strengths of association between two variables was measured using Pearson's correlation unless otherwise indicated. The degree ($k$) of a node $i$ is defined as the number of edges linked to $i$ (Fig. 1). Betweenness centrality ($C_b$) was computed as follows:

$$C_b(i) = \sum_{s \neq i \neq t} \frac{\sigma_{st}(i)}{\sigma_{st}}$$

where $s$ and $t$ are nodes in the network different from $i$, $\sigma_{st}$ denotes the number of

shortest paths from $s$ to $t$, and $\sigma_{st}(i)$ is the number of shortest paths from $s$ to $t$ that $i$ lies on. The network global topological efficiency was computed as the average inverse shortest paths between all vertices in the network as described in ref. 24.

Network attacks were simulated using the package igraph in R. We evaluated three different attack strategies on host specific networks: (i) a random attack where nodes are removed randomly; (ii) a centrality-based attack where nodes are removed deterministically based on the betweenness centrality scores and (iii) a pathogen-based attack where nodes are removed deterministically according to their centrality in the host–pathogen interactome.

**Function enrichment.** The functional enrichment (GO biological function) was calculated using David[25]. A functional enrichment was considered significant when the adjusted _P_-value < 0.05 (Benjamini-Hochberg correction). GO categories were grouped using REVIGO[26] and plotted using Cytoscape.

**Data availability.** The data that support the findings of this study are available from the corresponding authors upon reasonable request.

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

## Acknowledgements

M.T.B. would like to acknowledge support from the Programa Ramón y Cajal (RYC-2012-09999). This study has been funded by the Ministerio de Economía y Competitividad (SAF2014-56568-R) and a Research Grant 2016 by the European Society of Clinical Microbiology and Infectious Diseases (ESCMID) both to M.T.B. N.S.d.G acknowledges support of the Spanish Ministry of Economy and Competitiveness, 'Centro de Excelencia Severo Ochoa 2013-2017' and CERCA Programme from the Generalitat de Catalunya.

## Author contributions

M.T.B. and N.S.d.G. conceived and designed the experiments, N.C.A. and E.M.G. conducted the experiments, M.T.B. and N.S.d.G. wrote the manuscript. All authors read and approved the final text.

## Additional information

**Competing financial interests:** The authors declare no competing financial interests.

**Publisher's note**: 

