## [Peer Review File · Nature Communications]

Reviewers' comments:

Reviewer #1 (Remarks to the Author):

In this study, the authors provided correlations of the network topological information with pathogen fitness data in a comparative manner for bacteria and host-bacteria interactomes, to gain insights on the centrality-lethality relationships for bacterial proteins.

Abstract

- Organisms used in the study should be mentioned in the Abstract.
- Main conclusion is included at the end of the abstract, but the main results are missing. Before the last sentence of the abstract, main results of the comparison of bacteria and host-bacteria interactomes should be mentioned.

Introduction

- The pathogenic bacteria *Yersinia pestis* was selected for this study, probably because of available large amount of experimental *Y. pestis*-human PPI data. However, this reason is missing; it should be included in the Introduction.
- In the Introduction section, previous publications only about centrality-lethality rule are referred. However, centrality of the pathogen proteins in the host-pathogen interactome and also centrality of the pathogen-targeted host proteins (connectivity of the host proteins within the host's own intranetwork of PPIs) are two of the most intensely studied phenomena in the host-pathogen interaction research. Therefore, some of these previous works should be referred in the introduction.

Results and Discussion

- In addition to *Y. pestis*, other pathogenic bacteria *S. enterica* and *A. baumannii* were used in the study even they have no experimental host-pathogen PPI data. Those data were obtained by homology search. Why were these bacteria selected?
- "Hence, we reasoned that the critical impact of these proteins in pathogen survival might be related to the functions of their interacting partners. First, we examined the host proteins that interact with pathogen proteins relevant for infection (red data points). We found a significant enrichment in functions related to the immune response, transcription regulation and vesicle transport"

There are many previous studies which performed functional analysis of the host proteins that interact with pathogen proteins, and the critical impact of the pathogen proteins (with high connectivity values in the host-pathogen interactomes) was related to the functions of their interacting partners. These are the example studies with very similar findings (a significant enrichment in functions related to the immune response, transcription regulation and vesicle transport) for pathogenic bacteria: Dyer et al., 2008, PLoS Pathog. 4, 2, e32.; Dyer et al., 2010, PLoS One, 5, e12089; Yang et al., 2011 Infect. Immun., 79, 4413-4424; Durmuş et al. 2012 Front. Microbiol., 3, 46

- "Overall, our results demonstrate that the host-pathogen interactome is a complex self-contained network of interactions dominated by relevant hubs with the greatest contributions to infection fitness. These observations have important implications in drug design (Zoraghi and Reiner, 2013, Curr Opin Microbiol 16, 566-572). Antimicrobial development has focused on essential proteins (mainly enzymes) required for the pathogen to survive in culture (i.e., in vitro). Although this approach has been successful in the past, recent antimicrobial resistance threatens our capacity to develop new drugs. Our results indicate that strategic protein-protein interactions in the host-pathogen interactome should be explored as putative drug targets that may lay the foundation of a new class of antimicrobials."

Selecting highly connected pathogen proteins (within the host-pathogen interaction networks) as next generation antimicrobial/antiviral drug targets are proposed repeatedly as an efficient strategy to fight against with drug resistance. Additionally, hub proteins within intranetwork of pathogens and highly targeted (by pathogens within host-pathogen interaction networks) host proteins are also proposed as such good drug targets. In addition to Zoraghi and Reiner, (2013), there are many recent review papers, collecting such studies comprehensively (de Chassey et al., 2014, Genome Medicine, 6, 115; Pan et al. 2015, Briefings in Bioinformatics, doi: 10.1093/bib/bbv059; Durmuş et al., 2015, Frontiers in Microbiology, 6, 235).

Experimental Procedures

- Even though the title of the Methods section is "Experimental Procedures", no experiment was performed in the current study, as far as I understand. The fitness data for the pathogens were obtained from the literature.

- Why were only function terms considered in GO enrichment analysis? Biological process terms for the pathogen-targeted host proteins are also important to enlighten the infection strategies of the pathogens (i.e. Which host processes are attacked by pathogens?).

Reviewer #2 (Remarks to the Author):

The manuscript describes a topological and functional network analysis of the pathogen and host-pathogen interactome networks of *Yersinia pestis*, which are supported by the analyses of inferred interactions for two additional pathogens. The authors describe a correlation between the fitness effect of a gene deletion on growth in vitro and in vivo versus the degree of the encoded proteins in the pathogen and host-pathogen interaction network, respectively. In an analysis of functional annotations no enrichment for any function of the pathogen proteins that show an increased or decreased fitness in vivo compared to in vitro could be identified which motivated the authors to look at the functional annotations of the interacting host proteins. Here it was observed that proteins in functional categories related to immune function transcription and golgi transport are partners of proteins whose deletion is most detrimental during infection, whereas proteins with functions in cell metabolism and DNA homeostasis are partners of pathogen proteins whose deletion has the biggest effect during growth in vitro.

The manuscript is overall well written and addresses an important and very exciting question. At the same time, the manuscript leaves out important and central information and this information is critical to assess the validity of the conclusions.

One important area that requires much additional discussion is the nature and potential biases in the underlying datasets. Simply stating the database or publication from which the data were obtained is insufficient as the experimental methods and the underlying experimental design often give rise to biases. Such biases cannot be completely eliminated and are not a problem per se; especially not one that would prevent publication of the dataset in an original publication. However, at the same time some analyses may be severely affected by biases, which makes it imperative to reflect and test how they may affect conclusions - this is especially the case for network analyses such as the one at hand. To illustrate the point: the original centrality-lethality observation by Jeong et al. was made in the interactome of *S. cerevisiae*, for which two Y2H based interactome datasets were available. In the experiments producing one of these, ~130 proteins considered to be 'biologically important' were screened against most other of the 6000 yeast proteins in a matrix experiment (Uetz et al., Nature 2000). Naturally, for the ~130 proteins many more interaction partners were identified than for those proteins that were not screened as comprehensively, i.e. the 130 had a higher degree in the network. Problematically, an important piece of information to select 'important' proteins was the essentiality of the encoded gene. Thus, because the essential genes were screened more comprehensively they had a higher degree in the network thereby giving rise to the lethality-centrality correlation. A later study by Yu et al. (Science 2008) revealed that this correlation disappears when only systematic data are analyzed and that in such systematically acquired data the degree of a protein is correlated with the number of phenotypes observed in different conditions, thus revealing a centrality-pleiotropy correlation instead.

Important biases to consider in the manuscript at hand include methodological biases (e.g. mix of direct and indirect interactions in affinity-purification mass spec derived data), inspection bias in small-scale hypotheses driven data, or biased experimental design as in the aforementioned example. Thus, even though the authors rely on previously published datasets a discussion of the structure of these data and potential effects on the here presented conclusions is central to assess

the conclusions. This is the case for the initial network observations as well as the functional analyses. Of particular interest for the latter is whether an inspection bias could be partly responsible for the functional enrichment of immune and transcription related functions as highly connected host proteins in the host-pathogen interactome.

Another important question to briefly comment on is the nature of the host-pathogen interaction network, in particular what kind of pathogen proteins are interacting with which host-proteins and what is the significance and plausibility of such interactions. For example it can easily be seen how host receptors are interacting with pathogen proteins exposed on the extracellular surface of the microbe. For interactions among intracellular proteins this is less straightforward as only virulence effector proteins are delivered into the host cytosol whereas all other proteins of the pathogen are physically separated from host cell proteins. This is another question about the underlying data which need to be described in this computational analysis paper. Due to the rapidly evolving nature of the virulence effectors, which makes it generally very challenging to identify orthologues even in related pathogen species, another question is how host-pathogen interactions involving cytosolic host proteins can be inferred for other pathogens.

Related to the last point, it is unclear why pathogen proteins that are predominantly critical during in vitro proliferation would interact with host proteins that are enriched in functions during proliferation (Fig. 2D). Do these proteins ever see each other? What are examples of such pathogen proteins and why would they interact with the host DNA homeostasis machinery?

For the analysis shown in Fig 2B it would be important to see how the degrees of proteins in the different groups behave across a range of thresholds, i.e. is this a robust effect across a wider range or a local effect of the selected threshold.

Point by point response for NCOMMS-16-14494

Reviewer #1

We would like to thank the reviewer for his/her thorough revision and we greatly appreciate his/her comments and suggestions to improve the manuscript. We hope that the changes made will fulfil his/her expectations.

“Organisms used in the study should be mentioned in the Abstract”

“Main conclusion is included at the end of the abstract, but the main results are missing. Before the last sentence of the abstract, main results of the comparison of bacteria and host-bacteria interactomes should be mentioned”

We thank the reviewer for his/her comments on improving the abstract. We have now included a reference on the organisms studied and the main results on the comparison on interactomes are highlighted (Page 2, Lines 5-11).

The pathogenic bacteria Yersinia pestis was selected for this study, probably because of available large amount of experimental Y. pestis-human PPI data. However, this reason is missing; it should be included in the Introduction.

As rightly pointed by the reviewer, we found that *Y. pestis* was the only organism for which we have extensive information on host-pathogen interactome and also protein fitness measurements both *in vivo* and *in vitro*. We have now added a sentence in the Methods section discussing this (Page 5, Lines 8-11).

In the Introduction section, previous publications only about centrality-lethality rule are referred. However, centrality of the pathogen proteins in the host-pathogen interactome and also centrality of the pathogen-targeted host proteins (connectivity of the host proteins within the host's own intranetwork of PPIs) are two of the most intensely studied phenomena in the host-pathogen interaction research. Therefore, some of these previous works should be referred in the introduction.

We thank the reviewer for his/her suggestion. Accordingly, we have added a new paragraph in the introduction where we mention several studies on the centrality in pathogen and host-pathogen networks (Page 2, Lines 14-18)

In addition to Y. pestis, other pathogenic bacteria S. enterica and A. baumannii were used in the study even they have no experimental host-pathogen PPI data. Those data were obtained by homology search. Why were these bacteria selected?

These bacteria were selected because datasets on its protein fitness during infection were available (Wang et al. 2014; Chaudhuri et al. 2009). We have added a new sentence in the Methods section stating this fact (Page 5, Lines 10-11).

There are many previous studies which performed functional analysis of the host proteins that interact with pathogen proteins, and the critical impact of the pathogen proteins (with high connectivity values in the host-pathogen interactomes) was related to the functions of their interacting partners. These are the example studies with very similar findings (a significant enrichment in functions related to the immune response, transcription regulation and vesicle transport) for pathogenic bacteria: Dyer et al., 2008, PLoS Pathog. 4, 2, e32.; Dyer et al., 2010, PLoS One, 5, e12089; Yang et al., 2011 Infect. Immun., 79, 4413-4424; Durmuş et al. 2012 Front. Microbiol., 3, 46

We thank the reviewer for his/her insights regarding the functional analysis. We acknowledge that previous studies, including the ones cited by the reviewer, performed GO enrichment tests in the whole set of host proteins targeted by pathogens. However, we would like to notice that our calculations were not done for all the set of host interactors but for a particular group of proteins that we found to be of special interest.

One of our key observations in the paper is that some proteins essential for growth *in vivo* during infection are dispensable *in vitro* and that these proteins significantly make more interactions with the host (Fig. 2A,B; red group). We found that, **for this specific group of proteins**, there is a significant functional enrichment **when compared to the list of all known host proteins targeted by pathogens** (Fig. 2C). Hence, this is not a replication of a previously reported conclusion but a significant new finding that correlates protein essentiality during infection to its degree of interaction with host proteins, particularly in immune response and vesicle transport functions.

We acknowledge that this was probably not explained with enough clarity in the paper and we have expanded Fig. 2 legend accordingly (Page 9, Lines 24-26). Finally, we have also added the references mentioned by the reviewer in the Introduction (Page 2, Lines 14-18)

Selecting highly connected pathogen proteins (within the host-pathogen interaction networks) as next generation antimicrobial/antiviral drug targets are proposed repeatedly as an efficient strategy to fight against with drug resistance. Additionally, hub proteins within intranetwork of pathogens and highly targeted (by pathogens within host-pathogen interaction networks) host proteins are also proposed as such good drug targets. In addition to Zoraghi and Reiner, (2013), there are many recent review papers, collecting such studies comprehensively (de Chasseay et al., 2014, Genome Medicine, 6, 115; Pan et al. 2015, Briefings in Bioinformatics, doi: 10.1093/bib/bbv059; Durmuş et al., 2015, Frontiers in Microbiology, 6, 235).

As noted by the reviewer, some studies have suggested that hubs in host-pathogen networks might be interesting for drug design. However, a fundamental question remained unanswered: *Are host-pathogen protein-protein interactions associated to pathogen fitness during infection?*

In our paper, we demonstrate that there is a correlation between fitness and connectivity in the host-pathogen interactome. Also, we show that pathogen proteins that cause an extensive rewiring of host networks have a higher impact on fitness. To the best of our knowledge, a direct connection between protein connectivity in the pathogen-host interactome and its effect on fitness has been never reported before. Hence, we believe this is a landmark observation that strongly supports the hypothesis that targeting hubs in the host-pathogen interactome can lead to the discovery of new antimicrobials.

We acknowledge that this message may not be clearly stated in the conclusions section of our manuscript and we have added a line to highlight the importance of our findings (Page 4, Lines 35-36). Also, we have added the papers suggested by the reviewer as references in the Conclusions section (Page 4, Line 37)

Even though the title of the Methods section is "Experimental Procedures", no experiment was performed in the current study, as far as I understand. The fitness data for the pathogens were obtained from the literature.

We agree with the reviewer and we have changed the title accordingly to "Methods".

Why were only function terms considered in GO enrichment analysis? Biological process terms for the pathogen-targeted host proteins are also important to enlighten the infection strategies of the pathogens (i.e. Which host processes are attacked by pathogens?).

We thank the reviewer for his/her suggestion. When we performed GO enrichment analysis, we tested both molecular function and biological process, however GO terms with highest rank in biological process did not pass the significance test (χ^2 test with Benjamini-Hochberg correction for multiple hypothesis testing). This may be related to how GO domains are defined. While molecular function terms describe activities (including protein binding), biological process ontology terms integrate different molecular events. In the case of interactome analyses, which include physical interactions between proteins, it may be more reasonable to find significant enrichments with molecular functions rather than processes.

Reviewer #2

The manuscript is overall well written and addresses an important and very exciting question.

We would like to thank the reviewer for his/her enthusiasm on our work. We are also grateful for the extremely thoughtful and constructive suggestions to improve the manuscript. We believe that addressing the issues raised by the reviewer has significantly strengthened the conclusions presented in our paper and we hope that the revisions we propose will meet his/her expectations.

*"One important area that requires much additional discussion is the nature and potential biases in the underlying datasets. (...) To illustrate the point: the original centrality-lethality observation by Jeong et al. was made in the interactome of *S. cerevisiae*, for which two Y2H based interactome datasets were available. In the experiments producing one of these, ~130 proteins*

considered to be 'biologically important' were screened against most other of the 6000 yeast proteins in a matrix experiment (Uetz et al., Nature 2000). Naturally, for the ~130 proteins many more interaction partners were identified than for those proteins that were not screened as comprehensively, i.e. the 130 had a higher degree in the network. (...) Thus, because the essential genes were screened more comprehensively they had a higher degree in the network thereby giving rise to the lethality-centrality correlation. A later study by Yu et al. (Science 2008) revealed that this correlation disappears when only systematic data are analyzed and that in such systematically acquired data the degree of a protein is correlated with the number of phenotypes observed in different conditions, thus revealing a centrality-pleiotropy correlation instead. Important biases to consider in the manuscript at hand include methodological biases (e.g. mix of direct and indirect interactions in affinity-purification mass spec derived data), inspection bias in small-scale hypotheses driven data, or biased experimental design as in the aforementioned example. Thus, even though the authors rely on previously published datasets a discussion of the structure of these data and potential effects on the here presented conclusions is central to assess the conclusions.

We thank the reviewer for his/her detailed analysis on the nature of potential biases in the datasets used in our study. We acknowledge that such biases, especially in data derived from high-throughput analysis, have to be considered and controlled by all possible means. We would like to explain in the following lines all considerations and control calculations done to minimize the impact of potential biases. On the one side, we would like to explain why we consider that the bias related to an experimental design due to prior protein selection (a) and the use of small-scale data (b) do not apply to our study:

a. In contrast to the classic yeast two-hybrid (Y2H) screening, the dataset used was obtained by random Y2H meaning that no selection was made prior to the experiment about which proteins would be analysed. Contrary to the example provided by the reviewer in which the authors chose 130 proteins of interest, the proteins contained in the dataset used in our study were randomly picked after yeast mating and were never subjected to previous selection.

b. The dataset used in our study has 4.059 PPIs (>30 times the number of proteins analysed in the study of Uetz et al. and cited by the reviewer) and comes from the inspection of >500.000 diploid yeast cells, i.e. screened protein-protein interactions. Hence, in our opinion, the dataset should not be considered as a small-scale study. In any case, to reinforce the validity of our results, we have conducted bootstrapping analyses to provide a confidence interval in all r^2 values in Fig. 1. The results obtained further support our claim that centrality-lethality rule only holds when the host-pathogen interactome is considered.

On the other side, we acknowledge that all datasets have technical bias, which is inherent to any experimental analysis. Such bias cannot be removed but can be controlled by comparing independent studies. Therefore, in order to strengthen our results we have tested the correlation between the dataset used and the data published by Yang et al. (Infect and Immun; 2011) where the authors studied the protein-protein interaction network of *Y. pestis* virulence factors. We observed that the shared coverage between the two datasets was very low (as also mentioned by Yang et al. in their paper) and we could identify 12 proteins with information available in both datasets. Using this limited data, we asked whether proteins with highest impact on fitness have also a higher connectivity in Yang's dataset. Consistently with our previous results, we observed that proteins that were most essential in *Y. pestis* to infect the host (high fitness factor) had indeed a higher degree in the host-pathogen interactome (See Figure 1, $p=0.039$).

Figure 1. Average degree for proteins classified by fitness factor in the dataset of Yang et al. The number of observations was 6 in both groups.

In order to further study how technical biases in the interactome could affect our conclusions, we simulated random perturbation assays in the degree measurements and repeated the analysis. The noise was generated using the jitter function in R and the amount of noise (n) can be described by $n = f \frac{d}{5}$, where d is the smallest difference between values and f a factor defined by the user. We observed that, even for $f=25$ the conclusions remained sound and r^2 values for the centrality-lethality rule *in vitro* and *in vivo* were 0.82 ± 0.02 and 0.78 ± 0.02 , respectively.

Finally, we would like to point out that several studies (de Chasse et al. *Mol Sys Biol*, 2008; Zhang et al. *J Proteome Res*, 2009), including the one by Yang et al., have observed a strong agreement between Y2H and pull-down assays (an average of 70-85% of Y2H interactions could be validated by pull-down experiments), again supporting that the results are not biased.

All the results and discussion presented here has been included in the Supplementary Methods section (Page 2, Supplementary Information) under the title "*Analysis of potential biases in the definition of the centrality-lethality rule*".

This is the case for the initial network observations as well as the functional analyses. Of particular interest for the latter is whether an inspection bias could be partly responsible for the functional enrichment of immune and transcription related functions as highly connected host proteins in the host-pathogen interactome.

We thank the reviewer for his/her insightful analysis on functional enrichment. We would like to clarify that the enrichment analysis was not performed over the whole human proteome, which as stated by the reviewer, would generate a potential bias. Instead, the enrichment was performed over the list of host interactors in the host-pathogen interactome. We have included a sentence in the Fig. 2 legend to clarify this (Page 9, Lines 24-26).

Another important question to briefly comment on is the nature of the host-pathogen interaction network, in particular what kind of pathogen proteins are interacting with which host-proteins and what is the significance and plausibility of such interactions. For example it can easily be seen how host receptors are interacting with pathogen proteins exposed on the extracellular surface of the microbe. For interactions among intracellular proteins this is less straightforward as only virulence effector proteins are delivered into the host cytosol whereas all other proteins of the pathogen are physically separated from host cell proteins. This is another question about the underlying data which need to be described in this computational analysis paper. Due to the rapidly evolving nature of the virulence effectors, which makes it generally very challenging to identify orthologues even in related pathogen species, another question is how host-pathogen interactions involving cytosolic host proteins can be inferred for other pathogens.

We agree with the reviewer that the subcellular localization of pathogen proteins is important to understand whether the interaction predicted can take place during infection. While it would be extremely interesting to have localization data for pathogen proteins during infection, we are not aware of any available dataset with this information and such endeavour is far beyond the scope of our manuscript.

It is, in fact, a very difficult task as there are many possibilities for which pathogen proteins could localize in the host cytoplasm. Up to date, at least six different secretion systems have been discovered that can deliver pathogen proteins to the host cytosol (For a review in gram-negative secretion systems in bacteria see T.R.D. Costa et al. *Nat Rev Microb*, 2015). However,

many bacteria can also deliver proteins in outer membrane vesicles that fuse with the host membrane and deliver its content to the cell cytoplasm (C. Schwechheimer et al., *Nat Rev Microb*, 2015). Recent studies have been carried out to characterize the proteins and RNAs that are inside these vesicles (A. Elmi et al., *Infect. Immun.*, 2012) but this is a relatively new and unexplored area.

Related to the last point, it is unclear why pathogen proteins that are predominantly critical during in vitro proliferation would interact with host proteins that are enriched in functions during proliferation (Fig. 2D). Do these proteins ever see each other? What are examples of such pathogen proteins and why would they interact with the host DNA homeostasis machinery?

We thank the reviewer for his/her discussion on these results. We believe that there was a misunderstanding in Fig. 2D as the nodes do not represent host but pathogen proteins.

As described before, one of our key observations in the paper is that proteins essential for growth *in vivo* but dispensable *in vitro* have a higher connectivity in the host-pathogen interactome (Fig. 2A,B; red group) while proteins essential *in vitro* but dispensable *in vivo* display higher connectivity in the pathogen interactome (Fig. 2A,B; orange group). We therefore asked whether **the interactors** of the proteins in these two groups were enriched for defined functions. Hence, for the red group the enrichment was performed over the list of **host** interactors while in the orange group over the list of **pathogen** proteins.

We acknowledge that this message may not have been clearly stated in the figure legend and we are sorry if the text was confusing. In order to clarify this point, we completed Figure 2 legend (Page 9, Lines 24-26)

For the analysis shown in Fig 2B it would be important to see how the degrees of proteins in the different groups behave across a range of thresholds, i.e. is this a robust effect across a wider range or a local effect of the selected threshold

We thank the reviewer for his/her suggestion to improve our analysis in Fig. 2B. Accordingly, we have tested the results over a range of 10 different thresholds ($2 < \text{threshold} < 3$) and the results obtained were similar and displayed significant p-values (see **Fig. 2** in this point-by-point response). All results were found significant under $\alpha = 0.05$ except for some extreme threshold values where some comparisons were only significant under $\alpha = 0.10$. This is to be expected as extreme threshold values may shrink too much the number of observations in some groups. We have added a sentence in the manuscript explaining that the choice of the threshold does not invalidate the statistical significance of our results (Page 9, Lines 22-23).

Figure 2. Significance of p-values for comparisons shown in Fig 2B in the *Main Text*. Ten different thresholds were tested for all four comparisons in Fig. 2B (p₁, p₂, p₃, and p₄ represent the comparisons in the same order as appear in Fig. 2B). Results found statistically significant were coloured dark green ($\alpha = 0.05$) or light green ($\alpha = 0.10$).

Reviewer #1 (Remarks to the Author):

I can say that all of my comments have been satisfactorily addressed in the revised manuscript.

Reviewer #2 (Remarks to the Author):

The manuscript is greatly improved and most points have been addressed satisfactorily. A publication of the manuscript can be recommended after the following minor changes have been implemented.

Datasets:

even though the authors refer to previously published datasets, the nature of these datasets MUST be described briefly in the main text: what was screened, by which method, what fraction of bacterial and what fraction of human proteins was interrogated? Also, clearly indicate that for the other bacteria host-pathogen interactions were predicted and discuss biases due to this approach. In this context, also the caveat that for many interactions the biological plausibility may not be clear (not only relating to subcellular localization but more importantly also to the cellular localization - how do intracellular pathogen proteins interact with intracellular host proteins? must be discussed. This does not invalidate the findings, but it essential for an open discussion of their results.

Without experimental evidence the author should not state that pathogen proteins 'rewire' the network or that the network 'efficiency' decreases. These terms refer to specific effects associated with network dynamics about which no data are presented as only topological features are analyzed. Thus more neutral topological terms should be chosen that do not imply a particular but unsubstantiated type of perturbation.

Point-by-point response to reviewer's comments

Reviewer #2

Even though the authors refer to previously published datasets, the nature of these datasets MUST be described briefly in the main text: what was screened, by which method, what fraction of bacterial and what fraction of human proteins was interrogated? Also, clearly indicate that for the other bacteria host-pathogen interactions were predicted and discuss biases due to this approach.

We thank the reviewer for his/her suggestion; we have now added a description of all datasets used. In order not to break the flow of the main text with the dataset discussion, we have added this information in the Methods section. A description of the limitations in predicted host-pathogen interactions has been added to the Methods section.

In this context, also the caveat that for many interactions the biological plausibility may not be clear (not only relating to subcellular localization but more importantly also to the cellular localization - how do intracellular pathogen proteins interact with intracellular host proteins? must be discussed. This does not invalidate the findings, but it essential for an open discussion of their results.

We agree with the reviewer that a discussion about the biological plausibility of the interactions determined by high-throughput methodologies is appropriate and would help open discussion. Hence, we have added a new paragraph in the Discussion section.

Without experimental evidence the author should not state that pathogen proteins 'rewire' the network or that the network 'efficiency' decreases. These terms refer to specific effects associated with network dynamics about which no data are presented as only topological features are analyzed. Thus more neutral topological terms should be chosen that do not imply a particular but unsubstantiated type of perturbation.

We thank the reviewer for his/her comments. We have replaced the term “rewire” with more neutral terms (e.g. target). We could not change the term “efficiency” because this was the name given to this index by the authors who first defined it (see ref 21). However, in order to make clear that it refers to a topological and not a dynamic measurement, we have replaced the term “efficiency” by “topological efficiency”.